# Valorisation of *Prunus avium* L. By-Products: Phenolic Composition and Effect on Caco-2 Cells Viability

**DOI:** 10.3390/foods10061185

**Published:** 2021-05-25

**Authors:** Ana R. Nunes, Ana C. Gonçalves, Gilberto Alves, Amílcar Falcão, Cristina Garcia-Viguera, Diego A. Moreno, Luís R. Silva

**Affiliations:** 1CICS-UBI—Health Sciences Research Center, University of Beira Interior, 6200-506 Covilhã, Portugal; araqueln@gmail.com (A.R.N.); anacarolinagoncalves@sapo.pt (A.C.G.); gilberto@fcsaude.ubi.pt (G.A.); 2CNC—Centre for Neuroscience and Cell Biology, University of Coimbra, 3004-504 Coimbra, Portugal; 3Laboratory of Pharmacology, Faculty of Pharmacy, University of Coimbra, 3000-548 Coimbra, Portugal; acfalcao@ff.uc.pt; 4CIBIT—Coimbra Institute for Biomedical Imaging and Translational Research, University of Coimbra, 3000-548 Coimbra, Portugal; 5Phytochemistry and Healthy Foods Laboratory, Food Science and Technology Department, CEBAS-CSIC, 30100 Murcia, Spain; cgviguera@cebas.csic.es

**Keywords:** *Prunus avium* L., by-products, phenolic compounds, antiproliferative properties

## Abstract

*Prunus avium* L. by-products, such as stems, leaves, and flowers, are used in folk medicine to prevent and treat some diseases. However, their phenolic composition and in vitro bioactivities against tumor cells are poorly known. In this work, we compared the phenolic profile and the biological potential of aqueous infusions and hydroethanolic extracts of *P. avium* leaves, stems, and flowers from Saco cultivar, collected from the Fundão region (Portugal). Among the fifty-two phenolic compounds tentatively identified by HPLC-DAD-ESI/MS^n^, the hydroxycinnamic acids were the most abundant. Both extracts of stems revealed a higher activity against DPPH^•^. Meanwhile, hydroethanolic extracts from stems and flowers and aqueous infusions of flowers were the most effective in inhibiting the growth of the human epithelial colorectal adenocarcinoma (Caco-2) cells at concentrations above 200 μg/mL. More detailed knowledge about the phenolic composition and health-promoting properties of Portuguese *P. avium* by-products allows for increasing the biological and commercial value of these bio-wastes, which may have a positive impact on food and pharmaceutical industries, as on the valorization of the local economy.

## 1. Introduction

*Prunus avium* L. is a deciduous tree belonging to the Rosaceae family, Prunoideae subfamily, *Prunus* genus, and subspecies avium, which produces one of the most popularly appreciated red fruits, the sweet cherry [1]. This fruit possesses significant amounts of phenolic compounds, namely chlorogenic acids and anthocyanins, whose powerful antioxidant, anti-inflammatory, and chemopreventive properties are well-described [2,3,4,5,6,7].

Sweet cherry has been the focus of many studies due to its capacity to scavenge and neutralize free radicals, and consequently, attenuate or even mitigate the occurrence of oxidative stress-related diseases [4,7,8]. Nevertheless, tons of cherries are produced annually, resulting in large quantities of by-products as crop residues (e.g., leaves, stems, flowers, non-marketable fruits, and pits). The incorporation of these by-products in dietary supplements, functional foods, food fortification, food preservation, nutraceuticals, and pharmaceuticals has been encouraged as the cherry fruit and its by-products present significant bioactive compounds [9,10]. Few studies have already reported the richness of cherry leaves, stems, and flowers in hydroxycinnamic acids (e.g., 3- and 5-*O*-caffeoylquinic acids), flavan-3-ols (e.g., epigallocatechin and catechin), flavonols (e.g., quercetin 3-*O*-rutinoside, quercetin 3-*O*-glucoside, and kaempferol 3-*O*-rutinoside), and flavanones (naringenin and sakuranetin and their derivatives) [2,9,11] Moreover, their use can also be considered part of a strategy for sustainable fruit production, bringing numerous benefits for the regional circular economy and reducing agro-waste.

*P. avium* leaves and stems have been used since ancient times to boost the immune system due to their sedative, diuretic, anti-inflammatory, and draining properties [12,13]. On the other hand, their flowers are a symbol of renewal and vitality in Japanese culture [14].

In the present work, the identification of phenolic compounds from *P. avium* leaves, stems, and flowers of Saco cultivar using high-performance liquid chromatography (HPLC) coupled with mass spectrometer electrospray ionization mass (ESI-MS/MS) detection was performed. HPLC–DAD was used for the phenolic quantification of all samples. We also assessed their antioxidant capacity against 2,2-diphenyl-1-picrylhydrazil radical (DPPH^•^) and their effects on human epithelial colorectal adenocarcinoma (Caco-2) cells, herein reported for the first time.

## 2. Materials and Methods

### 2.1. Plant Material

*P. avium* leaves, stems, and flowers of the Saco cultivar (collected between April and June 2018) were provided by a local producer in Cimo da Aldeia—Pêro Viseu (40°12′36″ N 7°26′21″ W), Fundão region (Beira Interior, Portugal). After collection, the samples were transported to the CICS-UBI laboratory. The samples were frozen using liquid nitrogen, freeze-dried (ScanVac CoolSafe, LagoGene APS, Allerød, Denmark), and converted to a dried powder with a mean particle size lower than 910 μm. Posteriorly, they were stored at −20 °C and protected from light until analysis.

### 2.2. Aqueous Infusions and Hydroethanolic Extracts Preparation

Aqueous infusions of cherry by-products were prepared following a previously reported procedure [9]. Briefly, the dried powdered samples were subjected to infusion (1 g/100 mL) at 100 °C for 3 min. After that, they were filtered with a membrane filter (0.45 μm, Millipore, Bedford, MA, USA), frozen, freeze-dried, and kept at −20 °C in the dark until analysis. This protocol is similar to that involved in the infusions prepared commercially for human consumption. The obtained yields were 26.07 ± 1.22%, 28.42 ± 0.86%, and 18.40 ± 1.18% for leaves, stems, and flowers, respectively. On the other hand, and regarding the hydroethanolic extracts, 1 g of each sample was extracted with ethanol/water (50:50, *v/v*). The process involved 30 min of sonication, 2 h of maceration (at a temperature of 25 °C with agitation at 200 r/min), and another 30 min of sonication. The obtained extract was filtered (membrane with 0.45 μm, Millipore, Bedford, MA, USA), evaporated, and kept at −20 °C until further [9]. The extraction yields from dry material were 25.20 ± 1.90%, 28.89 ± 0.14%, and 22.31 ± 1.23% for leaves, stems, and flowers, respectively.

### 2.3. Total Phenolic Compounds Determination

The total phenolic compounds in *P. avium* leaves, stems, and flowers were estimated via a colorimetric assay, based on procedures described in [15] with some modifications. Briefly, 50 µL of the methanolic samples (2.5 mg/mL) was mixed with 450 µL of distilled water and 2.5 mL of 0.2 N Folin-Ciocalteu reagent. After 5 min of incubation at room temperature, 2 mL of a saturated sodium carbonate solution was added to the mixture, and the volume was adjusted to 5 mL with distilled water. Then, the reaction was kept in a water-bath at 30 °C for 90 min while protected from light. After that, the absorbance was read at 765 nm. Gallic acid was used to calculate the standard curve (50–500 mg/mL; y = 0.0011x + 0.0057; R^2^ = 0.9958). The experiments were performed in triplicate. The results were mean values ± standard deviations and expressed as molar concentration of gallic acid.

### 2.4. Total Flanonoids Determination

Flavonoid contents were assessed by the aluminum chloride colorimetric method [16]. Methanolic solutions of quercetin (purity ≥ 95%, 12.5–200 mg/L) were used to calculate the calibration curve (y = 0.0068x − 0.0135; R^2^ = 0.9969). The results were expressed in mg quercetin equivalents (QE) per gram of extract.

### 2.5. Phenolic Compounds Composition

#### 2.5.1. HPLC-DAD-ESI/MS^n^

Phenolics of leaves, stems, and flowers were analyzed by HPLC-DAD-ESI/MSn following the methodology previously described [5,17], with slight modifications. We used an Agilent HPLC 1100 series coupled to a mass detector ion trap spectrometer (model G2445A) equipped with an electrospray ionization interface and controlled by LC/MS software (Esquire Control Ver. 6.1. Build No. 534.1., Bruker Daltoniks GmbH, Bremen, Germany). The compounds were identified using their elution order, retention time, ultraviolet–visible spectra, fragmentation patterns, and diagnostic fragments in MS2 and MS3, with those obtained from standard compounds analyzed under the same conditions, and comparing the results obtained with the data reported in the literature [2,5,17,18].

#### 2.5.2. HPLC-DAD

Aqueous infusions and hydroethanolic extracts of leaves, stems, and flowers were dissolved in methanol (2 mg for leaves and flowers, and 10 mg for stems) and analyzed by HPLC-DAD (Agilent, Santa Clara, CA, USA), according to the method previously reported [3]. The quantification was performed by injecting 20 µL of each extract sample onto an LC model Agilent 1260 system (Agilent, Santa Clara, CA, USA), composed of an auto-sampler, coupled with a photodiode array detector, and the column used was a Nucleosil^®^ 100-5 C18 (Macherey-Nagel, Düren, Germany) [3]. The identification of the phenolics was completed through the absorbance recorded in the chromatograms relative to external standards at 280 nm for hydroxybenzoic acids, flavan-3-ols, flavanones, and flavones; 320 nm for hydroxycinnamic acids; 350 nm for flavanonols and flavonols; 500 nm for anthocyanins [5]. The phenolic compounds found in infusions and hydroethanolic extracts were identified by comparing their retention times and ultraviolet–visible absorption spectra in the 200–500 nm range with those of authentic standards. All the calibration curves of the external standards (concentration range of 1.5–100 µg/mL) used in the phenolics quantification are described in Table 1.

### 2.6. 2.2-Diphenyl-1-Picrylhydrazil Radical (DPPH•)-Scavenging Activity

A DPPH^•^-scavenging assay was evaluated following a reported procedure [9]. Briefly, seven different dilutions of infusion and hydroethanolic extracts of P. avium leaves, stems, and flowers were prepared in methanol, placed in a 96-well plate, and read at 515 nm, using a microplate reader Bio-Rad Xmark spectrophotometer. Three experiments were performed in triplicate. Ascorbic acid was used as a positive control.

### 2.7. Caco-2 Cell Culture and Treatment with By-products Extracts

Caco-2 cells were purchased from the American Type Culture Collection (ATCC; Manassas, VA, USA) and cultured as a monolayer in DMEM supplemented with 10% FBS and 1% penicillin/streptomycin (all from Sigma-Aldrich, MI, USA) at 37 °C in a humidified atmosphere of 95% air and 5% CO_2_. To determine the antiproliferative effects of infusions and hydroethanolic extracts of cherry by-products, cells were incubated for 24 h with different concentrations (50, 100, 200, 400, and 800 µg/mL) of these extracts, accordingly to the method previously described [4], with slight modifications. After 24 h of extract incubation, the 3-(4,5-dimethylthiazol-2-yl)-2,5-diphenyltetrazolium bromide (MTT) reduction and lactate dehydrogenase (LDH) release assays were performed.

#### 2.7.1. MTT Assay

Cell viability was assessing using the MTT assay following the protocol previously described [4]. The absorbance was measured at 570 nm using a microplate reader Bio-Rad Xmark spectrophotometer. Six independent experiments were performed in triplicate. Cell viability was measured as the percentage of absorbance compared to control.

#### 2.7.2. LDH Assay

LDH activity was spectrophotometrically evaluated at 340 nm, based on the conversion of pyruvate to lactate, using NADH as a cofactor, as previously described [4]. Six independent experiments were performed in triplicate.

### 2.8. Statistical Analysis

The results are expressed as mean values ± standard deviation (SD) and the experiments were carried out in triplicate. The results were analyzed using one-way ANOVA followed by Tukey’s test at a 95% level of significance. In the case of DPPH assay, the statistical comparison of all infusions and hydroethanolic extracts was performed using the Kruskal–Wallis test. Concerning the cellular-based assays, data from different extracts were compared by one-way ANOVA followed by Dunnett’s test as a post hoc test. Differences were considered statistically significant for *p* < 0.05 using Graphpad Prism Version 7.05 Software (San Diego, CA, USA).

## 3. Results and Discussion

### 3.1. Total Phenolics Content

Phenolic compounds constitute one of the major groups of phytochemicals, with a relevant antioxidant activity such as free radical-scavenging, hydrogen donation, singlet oxygen quenching, metal ion chelation, and also acting as a substrate for radicals such as hydroxyl and superoxide [19]. In order to make a general screening of the phenolics of *P. avium* leaves, stems, and flowers, the total phenols content was evaluated. The obtained results showed that *P. avium* stems presented a higher content of phenols (301.38 ± 5.9 mg GAE/g extract), followed by leaves and flowers (100.71 ± 8.30 and 81.20 ± 2.75 mg GAE/g extract, respectively (Table 2)). The obtained data are in line with those reported by [20], which found similar values in sweet cherry fruit. However, Serra et al. [21] reported higher values (1309 ± 12 mg GAE/100 g dw) in cherries from the Saco cultivar. In a study conducted by Prvulovic et al. [11], the total phenolics content in *P. avium* stems ranged from 12.96 to 31.85 mg GAE/g dry. These differences may be associated with some factors, such as the type of extraction used, the harvest region, and the climate, among others [22,23].

### 3.2. Total Flavonoids Content

Flavonoids are the main class of phenolics found in plants, and they are common components in the human diet. According to their chemical structure, they can be divided into flavonols, flavones, flavanones, isoflavones, flavan-3-ols, and anthocyanins [1]. These compounds have relevant biological activities, such as the ability to neutralize free radicals, and for that reason, their quantification in vegetal extracts is important [1]. The total flavonoids content was evaluated with the aluminum chloride colorimetric method (Table 3). As expected, the hydroethanolic extracts of P. avium by-products present higher concentrations of flavonoids than the respective infusions. The results showed that leaf hydroethanolic extract presented a higher content of flavonoids (35.17± 5.9 mg QE/g extract), followed by hydroethanolic extracts of flowers and stems (24.62 ± 0.39 and 15.25 ± 1.18 mg QE/g extract, respectively) (Table 3). Based on the obtained values for the total flavonoids content, it is possible to verify that the highest concentration of these compounds is in the extracts obtained using a solvent of moderate polarity (ethanol). Recently, similar results were found in P. avium by-products, such as stalk, pulp, seed, and leaf [24].

### 3.3. Analysis of Phenolic Compounds.

The individual phenolics present in *P. avium* leaves, stems, and flowers were tentatively identified based on a comparison of their retention times, ultraviolet–visible spectra, and the interpretation of their fragmentation patterns as obtained from MS^n^ spectra and by their relationship with other data reported in the literature. This study identified a total of fifty-two phenolic compounds, some of which were previously described in *P. avium* fruit [4,5,7] and its by-products [2,9]. The analysis allowed the identification of one hydroxybenzoic acid, twenty-five hydroxycinnamic acids, ten flavonols, five flavan-3-ols, seven flavanones, one flavanonol, one flavone, one anthocyanin, and one unknown compound (Table 4 and Table 5). As far as we know, this is the first study that characterizes the phenolic composition of leaves and flowers of sweet cherry, and the first concerned with the characterization of Saco variety stems from the Fundão region (Portugal), as evaluated by HPLC-DAD-ESI/MS^n^.

#### 3.3.1. Hydroxybenzoic Acids

In this work, one hydroxybenzoic acid has been described for the first time in the leaves and stems *of P. avium*. One signal (peak 5) at m/z 153 was detected and eluted at 15.7 min (Table 4). This compound was identified as a protocatechuic acid aglycone that corresponds to a loss of 162 Da (hexose moiety). This phenolic acid had already been reported in cherry fruits [5,6].

#### 3.3.2. Hydroxycinnamic Acids

A total of twenty-five hydroxycinnamic acids (peaks 1–4, 6–10, 12, 14–17, 20–21, 28, 33, 35, 38–39, 42, 44, 46–47) were identified in *P. avium* by-products (Table 4 and Table 5) using previously developed studies [2,6,9,25,26].

Peaks 1, 3, and 20 were identified as ferulic acids since they showed a fragmentation pattern at m/z 193 [ferulic acid-H]- after the loss of a hexosyl moiety (−162 mu), and the presence of a feruloyl fragment [25]. Bastos et al. [2] also detected their presence in *P. avium* stem extracts, infusions, and decoctions. On the other hand, these compounds have also been described in sweet cherry fruits [5,6]. Feruloyl di-hexose (peak 1) was not quantified in all infusions and hydroethanolic extracts (Table 5). Feruloyl hexose (peak 3) and feruloylquinic acid (peak 20) were not detected in leaves and stems, and it was not possible to quantify them in the flower infusion and hydroethanolic extract (Table 5).

Twelve signals (peaks 2, 4, 6–7, 9–10, 15, 21, 33, 35, 38, and 39) were detected that eluted between 13.7 and 31.3 min (Table 4). These components were identified as caffeoylquinic acids, and they have a fragmentation pattern at m/z 191 and/or at 179 [27]. Caffeoylquinic acid-glycoside (peak 7), 3-caffeoylquinic acid *cis* (peak 9), 3,5-dicaffeoylquinic acid 1 and 2 (peaks 10 and 35, respectively), 3,4-dicaffeoylquinic acid (peak 33), 5-caffeoylquinic acid *trans* (peak 21), and 4,5-dicaffeoyquinic acid (peak 39) had already been reported by Martini et al. [6] in sweet cherry cultivars. Moreover, peaks 4 and 7 were also identified as caffeoylquinic acid glycosides (m/z 515) (Table 4 and Table 5), and both possess a caffeoyl glucoside moiety (m/z 341) [28]. Peak 17 corresponds to 4-caffeyolquinic acid (molecular ion m/z z 353) with a fragment pattern at m/z 173 [25]. As far as we know, this is the first report concerning the presence of 4-caffeyolquinic acid in *P. avium* leaf infusion (ca. 908.93 ± 91.95 µg/g of dw) and hydroethanolic extract (ca. 466.37 ± 19.7 µg/g of dw), as shown in Table 5. However, its presence has already been reported in sweet cherries [5,6,27].

Peaks 14 and 16 were identified as *p*-coumaroylquinic acid derivative and *p*-coumaric acid derivative, respectively. They exhibited a molecular ion at m/z 337 and fragmentation ions at m/z 191 (Table 4 and Table 5) [2,25,28]. *P*-Coumaroylquinic acid has already been described in *P. avium* stems [2]. Moreover, peak 16 was described as a *p*-coumaroylquinic acid derivative, showing fragmentation ions at m/z 337.

The phenolic compounds corresponding to the peaks 46 and 47 were classified as coumaroyl-caffeoylquinic acids (molecular ion at m/z 499) (Table 4 and Table 5). Nevertheless, they possess distinct fragment patterns due to dehydration and the loss of caffeoyl and *p*-coumaroyl residues [29]. 3-Coumaroyl-5-caffeoylquinic acid (peak 46) was identified only in sweet cherry fruits [6].

Peak 12 was identified as a caffeoyl hexose (m/z 341) with a fragmentation at m/z 179 (Table 4 and Table 5) [6]. Similarly, in another study, Bastos et al. [2] tentatively identified these compounds as caffeic acid and trans-caffeic acid hexoside, based on the same pseudo-molecular ion, ultraviolet spectra, and fragmentation pattern. Additionally, two caffeoyl hexose derivatives (m/z 341) were tentatively identified (peaks 38 and 42), based on similar fragmentation (Table 4 and Table 5).

In sum, regarding hydroxycinnamic acids, *P. avium* leaves proved to be the by-products richest in these types of compounds. Totals of 51345.69 and 57605.22 µg/g of dw were obtained for the infusion and hydroethanolic extract, respectively (Table 5). Trans-5-caffeoylquinic acid (peak 21) was the major hydroxycinnamic acid found in both extracts, followed by 3-caffeoylquinic acid cis (peak 9). Our results were similar to those obtained by Jesus et al. [9], who also reported that hydroxycinnamics were the main compounds in leaves, representing 75.3% and 63.7% of total phenolic compounds in the infusion and hydroethanolic extract, respectively.

#### 3.3.3. Flavonols

Ten flavonols were found in *P. avium* leaves, stems, and flowers (Table 4 and Table 5), according to previous studies [2,5,30]. In this study, four quercetins (peaks 25, 26, 32, and 49) were detected (molecular ion at m/z 789, 771, 625, 609, and 463), as was a quercetin ion at m/z 301 (Table 4 and Table 5). Quercetin 3-O-rutinoside (peak 49) was the only flavonol quantified in all infusions and hydroethanolic extracts of *P. avium* leaves, stems, and flowers (Table 5). The leaf infusion was the richest one in this flavonol (ca. 6175.93 ± 148.22 µg/g of dw, comprising about 17.08% of total phenolic compounds), followed by flower hydroethanolic extract (1823.94 ± 38.9 µg/g of dw, containing about 7.71% of total phenolic compounds) (Table 5). These results agree with other recent work [9] showing that quercetin 3-*O*-rutinoside was the major flavonol found in leaf extracts. According to the literature, flavonols are the main phenolics found in leaves, and quercetin 3-*O*-rutinoside and quercetin 3-*O*-glucoside were also found in *P. avium* stems [2]. On the other hand, peak 26 was tentatively identified as quercetin O-rutinoside-O-hexoside. This compound exhibits a molecular ion at m/z 771, and typical fragmentations at m/z 609, 463, and 301 (Table 4). The presence of this flavonol in *P. avium* by-products agrees with other previous work (Bastos et al., 2015). Additionally, peak 26 was identified as quercetin di-hexoside and was only quantified in the flower hydroethanolic extract (Table 4 and Table 5). Quercetin 3-*O*-hexoside (peak 32) was detected at some levels in the stem and flower infusions and hydroethanolic extracts (Table 4 and Table 5), comprising less than 6% of total phenolic compounds.

On the other hand, six kaempferols (peaks 23, 27, 30, 34, 36, and 40) were found, all of which displayed a kaempferol fragment at m/z 285 [28]. Kaempferol *O*-rutinoside-*O*-hexoside (peak 23) and kaempferol 3-glucoside (peak 36) were found for the first time by Bastos et al. [2] in *P. avium* stems. Our study, as we far as known, detected them for the first time in flowers and leaves (Table 4 and Table 5). Kaempferol 3-*O*-rutinoside (peak 34) was identified in leaf infusions (1298.58 ± 44.2 µg/g) (Table 4 and Table 5), its presence being identified in previous works [9].

#### 3.3.4. Flavan 3-ols

Five flavan 3-ols (peaks 11, 13, 18–19, and 22) (Table 4 and Table 5) already described in cherries and their by-products were identified [2,5,6,9]. Catechin hexoside (peak 11) was tentatively identified, exhibiting a molecular ion at m/z 451 and fragmentation ions at m/z 289 and 245 (Table 4). This catechin was found in leaves and stems of *P. avium* (Table 5).

Furthermore, four procyanidins were identified in our samples (peaks 13, 18, 19, and 22). Two of these compounds were tentatively classified as type-B procyanidin dimers at m/z 577 (peaks 13 and 22) (Table 4 and Table 5). The presence of a procyanidin is supported by the fragment patterns at m/z 425 as well as water elimination at m/z 407 [28]. Our team reported similar findings for sweet cherry composition, which comprised about 18.47% total non-colored phenolics [5]. Procyanidin dimer B type 1 was found in different sweet cherries [6], while procyanidin dimer B type 2 was quantified in infusion and hydroethanolic extracts of *P. avium* stems (7149.5 ± 510.5 and 8810.67 ± 592.2 µg/g of dw, respectively) (Table 4 and Table 5). Procyanidin tetramer (peak 18; m/z 1153 molecular ion) and procyanidin trimer (peak 19; m/z 865 molecular ion) were also identified in *P. avium* by-products. As far as we know, this is the first study assessing the presence of these compounds in the leaves and stems of sweet cherries.

#### 3.3.5. Flavanones

A total of seven flavanones, eluted between 31.0 and 39.1 min, were identified in *P. avium* by-products, including four sakuranetins (peaks 37, 47, 51 and 52), two naringenins (peaks 41 and 48), and one pinocembrin (peak 44) (Table 4 and Table 5). Two sakuranetin 5-*O*-hexoside derivatives (peaks 37 and 52) were found for the first time in all cherry by-products studied in this work (Table 4 and Table 5). Both compounds have a fragmentation at m/z 285. Sakuranetin 5*-O*-hexoside (peak 51) was also identified in the same sample, presenting an ion at m/z 447 and releasing a fragment ion at m/z 285. This compound was quantified in the aqueous infusion and hydroethanolic extract (ca. 265.89 ± 9.8 and 214.66 ± 10.6 µg/g of dw, respectively) of *P. avium* leaves (Table 4), and these results are in accord with other previous works [2,9]. In addition, sakuranetin *O*-pentosyl-hexoside (peak 48) was tentatively identified based on the same fragmentation previously described (Table 4 and Table 5), and was found in stems. Previous data have already reported the presence of sakuranin in sweet cherries [21,31,32].

Naringenin-7-*O*-hexoside (peak 41) and naringenin hexoside (peak 48) presented a similar pseudo molecular ion at m/z 433 and 443 (Table 4 and Table 5), respectively. They release a fragmentation ion at m/z 271 (loss of hexose group) [2,9]. Naringenin 7-*O*-hexoside was quantified in stems infusion and hydroethanolic extracts (ca.1482.67 ± 15.94 and 1940.77 ± 51.2 µg/g of dw, respectively) (Table 5), comprising about 12% of the total phenolic compounds present in cherry stems. This compound was already described in *P. avium* stems and leaves, ranging between 51.9 and 4036.2 µg/g of dw, agreeing with other previous works [9]. Naringenin hexoside was also reported in sweet cherries [2,6]. The compound detected in peak 44 was tentatively identified as pinocembrin *O*-pentosyl-hexoside at a molecular ion m/z 549, releasing fragments at m/z 255 and 234. According to the literature, these fragments might be related to pinocembrin, a phenolic compound described in wood from different *Prunus* species [33]. A study conducted by Bastos et al. [2] reported this flavanone in *P. avium* stems extracts.

#### 3.3.6. Flavanonols

One flavanonol was detected in *P. avium* stems, namely aromadendrin *O*-hexoside (peak 31), which was eluted to 26.5 min and presented a molecular ion m/z 449 (Table 4 and Table 5). This compound released a fragment ion from the loss of 162 mu (hexosyl moiety), and its existence has already been reported in *P. avium* woods [33]. In the stem hydroethanolic extract, the mean value found of aromadendrin *O*-hexoside was around 172.96 ± 12.8 µg/g of dw (Table 5). Similarly, Bastos et al. [2] also identified and quantified this compound in stem extracts, although at a lower amount.

#### 3.3.7. Flavone

Chrysin 7-*O*-hexoside (peak 50) (Table 4 and Table 5) was the only flavone tentatively identified in cherry stems, exhibiting a molecular ion at m/z 415 and fragments at m/z 253 (Table 4). The identification of this flavone was achieved by considering previous reports focusing on *P. avium* and *P. cerasus* barks’ phenolic determinations [33,34,35]. This compound’s presence has already been reported in a study developed by Bastos et al. [2] in *P. avium* stems extracts.

#### 3.3.8. Anthocyanins

Cyanidin 3-*O*-rutinoside was the only anthocyanin identified in *P. avium* leaves, and was identified for the first time in this study (Table 4 and Table 5). This compound presented a molecular ion at m/z 595 and fragments at m/z 449 and 287. This anthocyanin is the major one found in sweet cherries, comprising around 70 and 90% of their total phenolics and anthocyanins contents, respectively [5,6]. However, in this study, we were unable to quantify this compound by HPLC-DAD (Table 5).

### 3.4. 2,2-Diphenyl-1-Picrylhydrazil Radical (DPPH•)-Scavenging Activity

Th eDPPH assay is a colorimetric test generally used to evaluate the antioxidant potential of pure compounds or extracts. Thus, the general screening of the antioxidant potential of cherry leaf, stem, and flower aqueous infusions and hydroethanolic extracts was performed by evaluating the scavenging activity against DPPH. All extracts revealed a strong antioxidant activity (Table 6). The infusions and hydroethanolic extracts of stems were the most active (IC_50_ = 19.04 ± 0.31 and 28.41 ± 0.55 µg/mL, respectively) (Table 6). Even so, all extracts showed less activity than the ascorbic acid-positive control (IC_50_ = 11.06 ± 0.37 µg/mL). The obtained results agree with another study that reported an IC50 = 23.38 µg/mL for the stem infusion [36]. Comparatively to the literature’s data, our results showed that both extracts of cherry leaf had a higher activity than sweet cherry leaf crude methanol extracts [37]. Additionally, all samples’ infusions and hydroethanolic extracts proved to be less efficient than cherry fruit hydroethanolic extracts (IC_50_ ranged between 12.12 and 43.3 μg/mL of dried extract) [3]. The antioxidant capacity of vegetal plant parts is intimately linked to phenolic content. Particularly, positive correlations were already reported between this assay and total phenolic acids (r = 0.9767 and r = 0.9200, *p* < 0.05, respectively).

### 3.5. Antiproliferative Effect on Caco-2-Cells

As previously described, *P. avium* leaves, stems, and flowers are very rich in phenolic compounds, and several studies have already proved that these phytochemicals are the main ones responsible for their bioactive properties [3,4,6,9,21]. In a previous study [4], the authors characterized the phenolic profile of extracts of sweet cherry fruit and reported that the major concentrations (between 200 and 800 µg/mL) of the colored fraction extract had a significant cytotoxic effect on Caco-2 cells. In this context, the antiproliferative effects of cherry leaf, stem, and flower infusions and hydroethanolic extracts on Caco-2 cells were also studied.

Caco-2 cells were used, due to their relevance to the analysis of the action of food components in intestinal epithelium issue. Thus, cells were exposed to different concentrations of extracts (50, 100, 200, 400, and 800 µg/mL) for 24 h, and after which cell viability was assessed through MTT assay. As shown in Figure 1, when we compare the infusions and the hydroethanolic extracts of leaves, stems, and flowers, it is possible to verify that concentrations above 200 µg/mL reduced cell viability (Figure 1). The hydroethanolic extracts of stems and flowers (Figure 1D,F), and the infusions of flowers (Figure 1E), significantly reduced cell viability (IC_50_ = 328.74 ± 2.37, IC_50_ = 349.76 ± 0.60, and IC_50_ = 364.79 ± 1.83 µg/mL, respectively). Phenolic compounds may be involved in the loss of cellular viability in cancer cells. It is of note that flavonoids can act as antioxidants or pro-oxidants, interfering in the carcinogenesis process [38]. According to the literature, these compounds and other phenolics such as chlorogenic acids have cytotoxic effects on cancer cells, preventing their development [39].

The cell membrane damage of Caco-2 cells after the incubation of infusions and hydroethanolic extracts of *P. avium* by-products was measured by the release of LDH. The LDH leakage assay is based on a measurement of the activity of this cytoplasmic enzyme in the extracellular medium. When the cell membrane loses its integrity, the LDH is released, and this can be quantified in the supernatant. Our results showed a significant increase in this enzyme for concentrations above 50 µg/mL in the leaf infusion (Figure 1A), stem infusion (Figure 1C), and flower hydroethanolic extract (Figure 1F). Thus, according to previous studies [5,40], it is possible that the cell death in Caco-2 is triggered by apoptosis.

## 4. Conclusions

The present work is the first to report on the phenolic characterization of leaves, stems, and flowers of Saco sweet cherry from the Fundão region by HPLC-DAD-ESI/MS^n^, and the effects of these phenolics on Caco-2 cells viability.

The obtained results reveal that hydroxycinnamic acids and flavonol glycosides were the main phenolics found in the *P. avium* by-products studied. Moreover, these compounds may be considered the main contributors to the bioactive properties demonstrated. Concerning biological potential, stem hydroethanolic extract was the most effective against DPPH^•^, while the aqueous infusion and hydroethanolic extract of flowers were the most cytotoxic for Caco-2 cells. In conclusion, this study may provide useful knowledge about cherry by-products’ phenolic compositions, contributing to their biological valorization, and thus offering an excellent source of bioactive compounds to be used by pharmaceutical and food industries. Furthermore, the recovery and valorization of cherry by-products may be beneficial for the regional circular economy and contribute to the reuse of agro-waste.

## Figures and Tables

**Figure 1 foods-10-01185-f001:**
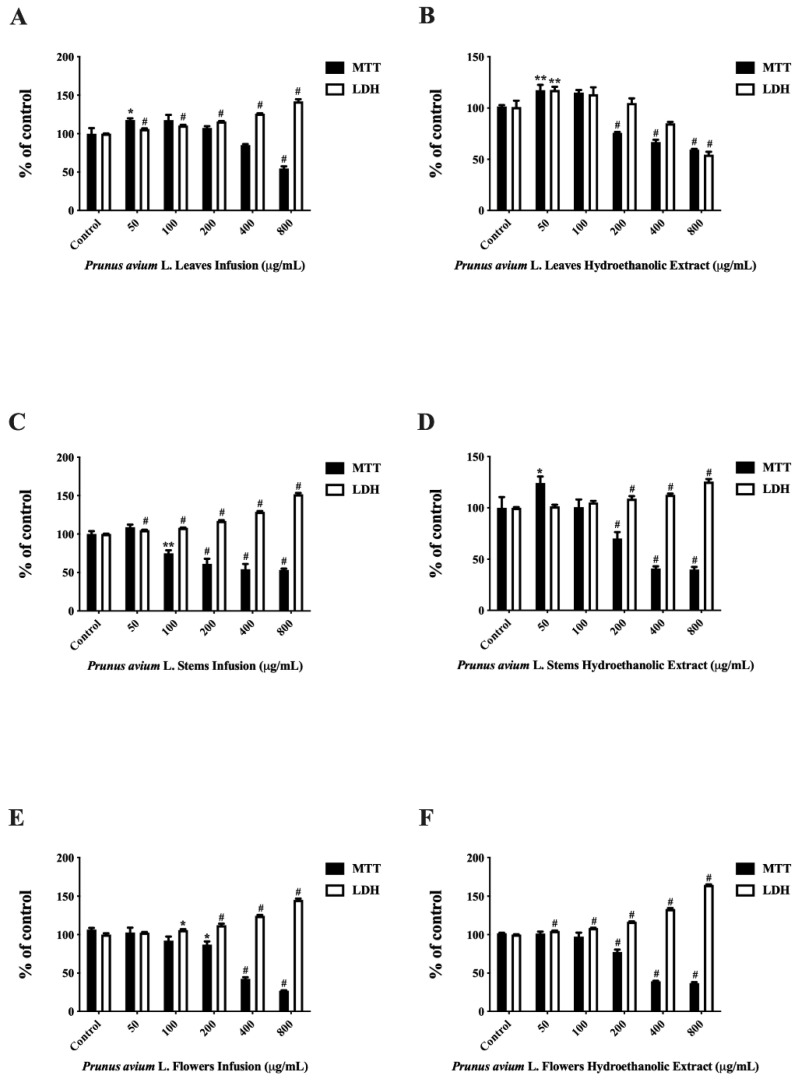
Effect of infusions and hydroethanolic extracts of *P. avium* leaves, stems, and flowers on Caco-2 cell line viability after 24 h of exposure, assessed by MTT reduction and LDH leakage assays. Values show mean ± SEM of at least six independent experiments performed in triplicate. (* *p* < 0.05, ** *p* < 0.01, and # *p* < 0.0001).

**Table 1 foods-10-01185-t001:** Calibration curves of external standards (concentrations range of 1.5–100 µg/mL) used in the quantification of phenolic compounds of leaves, stems, and flowers in *P. avium* extracts from the Fundão region (Portugal).

Phenolic Compound	Calibration Curve Equation	R^2^
Caffeic acid	y = 233.49x + 185.44	0.999
Catechin	y = 34.52x + 134.38	0.998
Chlorogenic acid	y = 46.37x + 170.34	0.999
Chrysin	y = 69.375x − 43.531	0.999
*p*-coumaric acid	y = 260.73x -137.54	0.999
Ferulic acid	y = 136.37x + 154.91	0.999
Gallic acid	y = 54.13x + 131.11	0.997
*p*-hydroxybenzoic acid	y = 36.18x + 121.90	0.995
Kaempferol 3-*O*-rutinoside	y = 41.442x + 184.11	0.999
Naringenin 7-*O*-glucoside	y = 41.039x + 183.36	0.999
Quercetin	y = 48.56x − 11.222	0.999

Feruloyl di-hexose (peak 1), feruloyl hexose (peak 3), and feruloylquinic acid (peak 20) were quantified as ferulic acid, while caffeoylquinic acid derivative 1 and 2 (peaks 2 and 6), caffeoylquinic acid glycoside-derivative (peak 4), caffeoylquinic acid-glycoside (peak 7), caffeoyl hexose derivative 1, 2 and 3 (peaks 8, 38 and 42), 3-caffeoylquinic acid cis and 5-caffeoylquinic acid trans (peak 9 and 21), 3,5-dicaffeyolquinic acid 1 and 2 (peaks 10 and 35), caffeoyl hexose (peak 12), dicaffeoylquinic acid 1 and 2 (peak 15 and 43), 4-caffeoylquinic acid (peak 17), 3,4-dicaffeyolquinic acid (peak 33), and 4,5-dicaffeoylquinic acid (peak 39) were quantified as chlorogenic acid. Protocatechuic acid-glycoside (peak 5) as p-hydroxybenzoic, while catechin hexoside (peak 11), procyanidin dimer B type 1 and 2 (peaks 13 and 22), procyanidin tetramer (peak 18), procyanidin trimer (peak 19), quercetin-3-O-rutinoside-O-hexoside (peak 25), quercetin di-hexoside (peak 26), quercetin-3-O-hexoside and quercetin-3-O-rutinoside (peaks 32 and 49) were quantified as quercetin. On the other hand, p-coumaroylquinic acid derivative (peak 14), p-coumaric acid derivative (peak 16), p-coumaroylquinic acid (peak 28), 3-coumaroyl-5-caffeoylquinic acid and 3-coumaroyl-4-caffeoylquinic acid (peaks 45 and 46) were quantified as p-coumaric acid. Kaempferol-O-rutinoside-O-hexoside and kaempferol-O-rutinoside-O-hexoside derivative (peaks 23 and 27), kaempferol-di-hexoside (peak 30), kaempferol-3-O-rutinoside (peak 34), kaempferol 3-hexoside (peak 36), and kaempferol-3-O-acteyl-hexoside (peak 40) were quantified as kaempferol 3-O-glucoside. Finally, sakuranetin-5-O-hexoside derivative 1 and 2 (peaks 37 and 52), sakuranetin-O-pentosylhexoside (peak 47), sakuranetin-5-O-hexoside (peak 51), pinocembrin-O-pentosylhexoside (peak 44), naringenin-7-O-hexoside, and naringenin hexoside (peaks 41 and 48) were quantified as naringenin-7-O-glucoside.

**Table 2 foods-10-01185-t002:** Total phenolic content of leaves, stems, and flowers in *P. avium* from the Fundão region (Portugal).

	Leaves	Stems	Flowers
Total Phenols Content (mg GAE per g dw)	100.71 ± 8.30	301.38 ± 5.91 ^a^	81.20 ± 2.75 ^a,b^

Values are expressed as mean ± standard deviation of three experiments. ^a^ Significant results (*p* < 0.05) are indicated as vs. leaves, ^b^ as vs. stems.

**Table 3 foods-10-01185-t003:** Total flavonoids content of leaves, stems, and flowers in *P. avium* infusion and hydroethanolic extracts from Fundão region (Portugal).

	Extracts	Total Flavonoids Content (mg QE per g dw)
**Leaves**	**Infusion**	31.63 ± 2.24
	**Hydroethanolic**	35.17 ± 2.62
**Stems**	**Infusion**	9.93 ± 1.19 ^a,b^
	**Hydroethanolic**	15.25 ± 1.18 ^a,b,c^
**Flowers**	**Infusion**	22.67 ± 0.73 ^a,b,c,d^
	**Hydroethanolic**	24.62 ± 0.39 ^a,b,c,d^

Values are expressed as mean ± standard deviation of three experiments. ^a^ Significant result (*p* < 0.05) is indicated as vs. leaves infusion, ^b^ as vs. leaves hydroethanolic, ^c^ s vs. stems infusion, ^d^ vs. stems hydroethanolic.

**Table 4 foods-10-01185-t004:** Retention time (Rt), wavelengths of maximum absorption in the ultraviolet-visible region (λmax), mass spectral data and identification of phenolic compounds of leaves, stems, and flowers in *P. avium* extracts from Fundão region (Portugal).

Peak	Phenolic Compounds	HPLC-DAD-ESI-MS^n^ Characteristics			
		R_t_ (min)	λ_max_ (nm)	Molecular Ion [M + H] (*m/z*)	Fragments MS/MS (*m/z*)
1	Feruloyl di-hexose	13.1	320	518	355/356, 337, 193, 176
2	Caffeoylquinic acid derivative 1	13.7	320	371	353, 191, 135
3	Feruloyl hexose	14.2	320	355	193, 176
4	Caffeoylquinic acid glycoside-derivative	15.3	320	515	341, 335, 179, 191
5	Protocatechuic acid-glycoside	15.7	280	315, dimer adduct 631	153
6	Caffeoylquinic acid derivative 2	16.1	350	503	341, 179
7	Caffeoylquinic acid-glycoside	16.4	320	515	341, 179
8	Caffeoyl hexose derivative 1	17.2	350	683	521, 529, 341, 315, 179
9	3-Caffeoylquinic acid *cis*	17.6	320	353, dimer adduct 707	191, 179, 135
10	3,5-diCaffeoylquinic acid 1	18.2	320	515	353, 191, 179, 135
11	Catechin hexoside	18.4	280	451	289, 245
12	Caffeoyl hexose	18.7	320	341	179, 135
13	Procyanidin dimer B type 1	19.2	280	577	289, 407, 425, 451
14	*p*-Coumaroylquinic acid derivative	19.6	320	679	337, 517, 337, 191, 162
15	diCaffeoylquinic acid 1	19.8	320	515	323, 191, 353
16	*p*-Coumaric acid derivative	20.1	320	337, dimer adduct 675	162,191
17	4-Caffeoylquinic acid	20.8	320	353, dimer adduct 707	173
18	Procyanidin tetramer	21.0	280	1153	865, 577, 289, 245
19	Procyanidin trimer	21.2	280	865	577, 289, 245
20	Feruloylquinic acid	21.2	320	367	193
21	5-Caffeoylquinic acid *trans*	21.5	320	353	191, 179
22	Procyanidin dimer B type 2	22.0	280	577	425, 289, 407, 451
23	Kaempferol *O*-rutinoside-*O*-hexoside	22.7	350	755	593, 285, 695
24	Cyanidin 3-*O*-rutinoside	23.3	280	595	449, 287
25	Quercetin 3-*O*-rutinoside-*O*-hexoside	24.4	350	771	609, 463, 301
26	Quercetin di-hexoside	25.0	350	625	453, 301
27	Kaempferol *O*-rutinoside-*O*-hexoside derivative	25.3	350	771	593, 285, 327
28	*p*-Coumaroylquinic acid	25.5	320	337, dimer adduct 675	337, 191
29	Unknown 1	25.5	360	465	285
30	Kaempferol-di-hexoside	25.6	350	609	285, 447
31	Aromandendrine *O-*hexoside	26.5	350	447	287
32	Quercetin-3-*O*-hexoside	28.0	350	463	301, 271, 179
33	3,4-diCaffeoylquinic acid	28.5	320	515	353, 173
34	Kaempferol 3-*O*-rutinoside	29.2	350	593	285, 256
35	3,5-diCaffeoylquinic acid 2	30.0	320	515	353, 191
36	Kaempferol 3-hexoside	30.7	350	447	285
37	Sakuranetin 5-*O*-hexoside derivative 1	31.0	280	447	285
38	Caffeoyl hexose derivative 2	31.2	320	503	341, 179
39	4,5-diCaffeoylquinic acid	31.3	320	515	353, 179, 173
40	Kaempferol 3-*O*-acetyl-hexoside	32.7	350	489	285
41	Naringenin 7-*O*-hexoside	33.0	280	433	271, 151, 313
42	Caffeoyl hexose derivative 3	33.8	320	869	451, 341
43	diCaffeoylquinic acid 2	34.7	320	515	353, 335, 191, 173
44	Pinocembrin-*O*- pentosylhexoside	35.0	280	549	255, 234
45	3-Coumaroyl-5-caffeoylquinic acid	35.0	320	499	337,163, 173
46	3-Coumaroyl-4-caffeoylquinic acid	35.3	320	499	337, 353, 173
47	Sakuranetin-*O*-pentosylhexoside	35.5	280	579	285, 270
48	Naringenin hexoside	35.9	280	443	271
49	Quercetin 3-*O*-rutinoside	36.9	350	609	301, 271
50	Chrysin 7-*O*-hexoside	37.0	280	415	253, 208
51	Sakuranetin 5-*O*-hexoside	38.3	280	447	285
52	Sakuranetin 5-*O*-hexoside derivative 2	39.1	280	593	447, 285

**Table 5 foods-10-01185-t005:** Quantification of phenolic compounds (µg/g of dried samples) of leaves, stems, and flowers in *P. avium* infusion and hydroethanolic extracts from the Fundão region (Portugal).

Peak	Phenolic Compounds	Leaves		Stems		Flowers	
		Infusion	Hydroethanolic	Infusion	Hydroethanolic	Infusion	Hydroethanolic
7	Caffeoylquinic acid-glycoside	nq	nq	1936.04 ± 18.9	2117.33 ± 25.4 ^c^	nd	nd
9	3-Caffeoylquinic acid *cis*	18,667.85 ± 162.2	20,215.87 ± 917.3 ^a^	196.23 ± 14.1 ^a,b^	320.23 ± 46.7 ^a,b^	23,294.66 ± 653.4 ^a,b,c,d^	15,996.99 ± 335.4 ^a,b,c,d,e^
10	3,5-diCaffeoylquinic acid 1	nq	nq	nq	158.85 ± 4.7	2948.25 ± 29.5 ^d^	1309.65 ± 47.9 ^d,e^
15	diCaffeoylquinic acid 1	2210.36 ± 99.3	3375.96 ± 98.8 ^a^	nd	nd	nq	nq
16	*p*-Coumaric acid derivative	1482.13 ± 14.9	1335.62 ± 51.9 ^a^	nd	nd	nd	nd
17	4-Caffeoylquinic acid	908.93 ± 81.95	466.37 ± 19.7 ^a^	nd	nd	nd	nd
21	5-Caffeoylquinic acid *trans*	24,425.04 ± 897.3	27,210.54 ± 1415.7 ^a^	1095.56 ± 167.7 ^a,b^	1338.68 ± 40.2 ^a,b^	3841.41 ± 304.04 ^a,b,c,d^	640.77 ± 28.6 ^a,b,c,d,e^
22	Procyanidin dimer B type 2	nq	nq	7149.5 ± 510.5	8810.67 ± 529.2 ^c^	nd	nd
23	Kaempferol-*O*-rutinoside-*O*-hexoside	nq	nq	nd	nd	5313.35 ± 91.7	2676.57 ± 134.9 ^e^
26	Quercetin di-hexoside	nd	nd	nd	nd	nq	348.22 ± 6.3
28	*p*-Coumaroylquinic acid	473.68 ± 6.3 ^a^	450.79 ± 9.1	nd	nd	nd	nd
31	Aromandrine *O-*hexoside	nd	nd	nq	172.96 ± 18.9	nd	nd
32	Quercetin 3-*O*-hexoside	nq	nq	665.76 ± 1.6	1025.78 ± 18.2 ^c^	702.74 ± 12.8 ^c,d^	555.16 ± 12.9 ^c,d^
34	Kaempferol 3-*O*-rutinoside	1298.58 ± 44.2	nq	nq	nq	nd	nd
36	Kaempferol 3-hexoside	nq	1542.19 ± 112.13	nd	nd	nq	nq
41	Naringenin 7-*O*-hexoside	nd	nd	1482.67 ± 15.94	1940.77 ± 51.2 ^c^	nd	nd
45	3-Coumaroyl-5-caffeoylquinic acid	696.45 ± 10.9	905.1 ± 11.9 ^a^	nd	nd	340.21 ± 0.95 ^a,b^	305.5 ± 36.2 ^a,b^
46	3-Coumaroyl-4-caffeoylquinic acid	2481.25 ± 51.8	3644.97 ± 64.00 ^a^	nd	nd	327.81 ± 21.1 ^a,b^	nq
48	Naringenin hexoside	326.60 ± 56.1	689.26 ± 58.5 ^c^	nd	nd	nd	nd
49	Quercetin 3-*O*-rutinoside	6175.93 ± 148.22	3653.48 ± 22.7 ^a^	404.39 ± 9.7 ^a,b^	767.00 ± 19.4 ^a,b,c^	2512.03 ± 6.03 ^a,b,c,d^	1823.94 ± 38.9 ^a,b,c,d,e^
51	Sakuranetin 5-*O*-hexoside	265.89 ± 9.8	214.66 ± 10.6 ^a^	nq	nq	nq	nq
	Σ	36,142.42	63,704.81	12,932.15	16,181.65	39,280.46	23,656.8

Values are expressed as mean ± standard deviation of three assays. Σ, sum of the determined phenolics; nd, not detected; nq, not quantified. ^a:^ Significant result (*p* < 0.05) is indicated as vs aqueous infusion leaves; ^b^: vs hydroethanolic leaves; ^c^: vs aqueous infusion flowers; ^d^: vs hydroethanolic flowers; ^e^: vs aqueous infusion stems.

**Table 6 foods-10-01185-t006:** IC_50_ (µg/mL) values of DPPH radical-scavenging activity of *P. avium* leaf, stem, and flower infusions and hydroethanolic extracts from Saco cultivar.

	Extracts	DPPH^•^
Leaves	Infusion	55.12 ± 1.11
	Hydroethanolic	51.52 ± 0.84
Stems	Infusion	28.41 ± 0.55
	Hydroethanolic	19.04 ± 0.31 ^a^
Flowers	Infusion	56.64 ± 0.91
	Hydroethanolic	194.1 ± 2.07 ^a^

Values are expressed as mean ± standard deviation of three experiments. ^a^ Significant result (*p* < 0.05) as vs. infusion leaves.

## Data Availability

Data are contained within this article.

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
