# Peer review of "Valorisation of Prunus avium L. By-Products: Phenolic Composition and Effect on Caco-2 Cells Viability"

_foods, 2021, doi:10.3390/foods10061185_

Round 1

Reviewer 1 Report

The article titled Valorisation of Prunus avium L. By-products: Phenolic Composition and Effect on Caco-2 Cells Viability seems to be very interesting. An article offering the possibility of using the by-products of a popular plant. The article is well written, the introduction contains important data taking into account the latest knowledge. The methodology was correctly described. However, the extraction conditions should be entered in more detail. The results and discussion of the research were presented in a very clear way, using the appropriate division into sections.

Author Response

The authors thank and agree with the reviewer on this comment. As suggested, the extraction conditions were entered in more detail in the manuscript. In this study, two different methods of extracts preparation were used: aqueous infusion and hydroethanolic extract. The aqueous infusion was used to mimetite the herbal preparations in our daily life. On the other hand, the hydroethanolic extract was chosen because it allows a better extraction of phenolic compounds, according to several reports in the literature. Most phytochemicals present in plants have in their composition molecules with phenolic structures, which are more easily extracted by polar mixtures, such as alcoholic solvents. (Please see now lines 73 to 82 of the revised version of the manuscript)

Reviewer 2 Report

In the present work, the identification and quantification of phenolic compounds from P. avium leaves, stems, and flowers of Saco cultivar was performed. Furthermore, their antioxidant capacity against the radical 2,2-diphenyl-1-picrylhydrazil (DPPH •) and their effects on human epithelial colorectal adenocarcinoma cells (Caco-2) was evaluated.

The work they have done is very interesting and the manuscript is well written and the English language also seems correct throughout most of the document.

However, there are some corrections to be made.

The scientific name P. avium must be written in italics. Check the entire text. eg: line 27, 274, 318, 338, 373, 413,

line 344: Write the scientific name in the abbreviated form

The name "Saco" is sometimes written in italics (eg line 369) and sometimes not. Choose how to write and unify all the text.

Line 370: Format the text of the table caption

in “material and methods”, the DPPH assay is missing. I think it is necessary to insert the method.

Author Response

The authors thank to the reviewer on this comment. All the suggestions were corrected throughout the manuscript, as suggested.

Reviewer 3 Report

the paper is well presented and the materials and methods are correctly reported. In addition to the quantification of the total phenols, (Tab.2) I also suggest a quantification of the total Flavonoids, with a colorimetric method. Table 4 is difficult to read, only quantifiable substances should be reported for clarity. I suggest, if possible, to add at least one other antioxidant activity test, for example ORAC; the DPPH test alone does not fully define the antioxidant capacity. 

Author Response

The authors thank the reviewer for this comment that contribute to enhance the quality and reading of the manuscript. We followed the reviewer’s suggestions and inserted the quantification of the total Flavonoids with a colorimetric method, including a new table of the results (Table 3), and the respective discussion of them. (Please see now lines 97 to 101, and 186 to 206 7of the revised version of the manuscript).

As suggested, Table 4 (now Table 5) was reorganized, leaving only the compounds possible of quantification (Please see now lines 379 to 381 of the revised version of the manuscript).

Regarding to add one other antioxidant activity test, we are in accordance with the reviewer’s suggestion. Unfortunately, the 

The authors thank the reviewer for this comment that contribute to enhance the quality and reading of the manuscript. We followed the reviewer’s suggestions and inserted the quantification of the total Flavonoids with a colorimetric method, including a new table of the results (Table 3), and the respective discussion of them. (Please see now lines 97 to 101, and 186 to 206 7of the revised version of the manuscript).

As suggested, Table 4 (now Table 5) was reorganized, leaving only the compounds possible of quantification (Please see now lines 379 to 381 of the revised version of the manuscript).

Regarding to add one other antioxidant activity test, we are in accordance with the reviewer’s suggestion. Unfortunately, the time between the receipt of “Major Revisions” and the date on which responses were sent was too short to rigorously carry out all requested tests. However, in future publications, we commit to insert other antioxidant tests that prove the antioxidant potential of Prunus avium by-products, such as ABTS, ORAC, and/or FRAP assays. Furthermore, it is possible to verify other recently published paper of our team that proved the antioxidant capacity of the samples used in this study:

-        Jesus F, Gonçalves AC, Alves G, Silva LR. Exploring the phenolic profile, antioxidant, antidiabetic, and anti-hemolytic potential of Prunus avium vegetal parts. Food Research International (2019) 116: 600-610.

Thank you for all suggestions.

Reviewer 4 Report

the paper is sufficiently well written even if it does not present any novelty from a methodological point of view, therefore the information provided suffers from this situation, different studies on the health potential of the extracts would have enhanced the work.

there are some minor error in the writing 

Author Response

The authors thank and agree with the reviewer on this comment. However, this study is a preliminary report about the phenolic composition and possible health-promoting properties of Prunus avium by-products of a traditional Portuguese cultivar. Although the paper does not present novelty from a methodological point of view, it provides important information, and the first time, about composition in phenolic content and biological properties of cherry by-products. According to the obtained results, the richness of these by-products was revealed and their potential incorporation and/or use in food and pharmaceutical preparations. Furthermore, the recovery and valorization of cherry by-products may be beneficial for the regional circular economy and contribute to the reuse of agro-waste.

As suggested, minor errors were corrected, and the English was revised.

Reviewer 5 Report

This manuscript reports the compared the phenolic profile and the biological potential of extracts of P. avium leaves, stems, and flowers, 52 phenolic compounds tentatively identified by HPLC-DAD-ESI/MSn and DPPH radical scavenging activity and inhibit effect of the Caco-2 cells evaluated. The manuscript need a minor revision.

  1. in Figure 1, “Extract” The capitalization of the beginning of the word needs to be unified.
  2. Line 79, rpm change to r/min
  3. Prunus avium in the title should be italicized.
  4. Line 87, equivalent concentration unit should be changed to molar concentration unit.

Author Response

In Figure 1, “Extract” The capitalization of the beginning of the word needs to be unified. The authors thank the reviewer for this comment. The term “Extract” in Figure 1 was corrected. (Please see now new Figure 1 of the revised version of the manuscript).

Line 79, rpm change to r/min. The term “rpm” was replace for “r/min”. (Please see now new line 82 of the revised version of the manuscript).

Prunus avium in the title should be italicized. The term “Prunus avium” was placed in italics. (Please see now new line 82 of the revised version of the manuscript).

Line 87, equivalent concentration unit should be changed to molar concentration unit. The authors thank the reviewer suggestion. In fact, the equivalent concentration could be changed to molar concentration unit. However, in this assay, and according to several reports in the literature (see list of references above), the better unit to present the results is effectively “mg of gallic acid equivalents (GAE) per gram of extract”. Based on that information and considering the other papers with the same quantification, we decided to pursue this strategy in our study.

Coimbra AT, Luís AFS, Batista MT, Ferreira SM, Duarte AP. Phytochemical Characterization, Bioactivities Evaluation and Synergistic Effect of Arbutus unedo and Crataegus monogyna Extracts with 

  • Amphotericin B. Current Microbiology (2020) 77: 2143-2154.
  • Serra, AT, Duarte RO, Bronze MR, Duarte CMM. Identification of bioactive response in traditional cherries from Portugal. Food Chemistry (2011) 125: 318-325.
  • Barreira, JCM, Ferreira ICFR, Oliveira MBPP, Pereira JA. Antioxidant activity and bioactive compounds of ten Portuguese regional and commercial almond cultivars. Food and Chemical Toxicology (2008) 46: 2230-2235.

Barreira, JCM, Ferreira ICFR, Oliveira MBPP, Pereira JA. Antioxidant activities of the extracts from chestnut flower, leaf, skins, and fruit. Food Chemistry (2008) 107: 1106-1113.